# The Correlation between Islet β Cell Secretion Function and Gallbladder Stone Disease: A Retrospective Study Based on Chinese Patients with Newly Diagnosed Type 2 Diabetes Mellitus

**DOI:** 10.3390/biomedicines11102840

**Published:** 2023-10-19

**Authors:** Tiantian Wu, Qiang Wang, Changsheng Pu, Keming Zhang

**Affiliations:** Department of Hepatobiliary Surgery, Peking University International Hospital, Beijing 100001, China; wangqiang1@pkuih.edu.cn (Q.W.); puchangsheng@pkuih.edu.cn (C.P.); zhangkeming@pkuih.edu.cn (K.Z.)

**Keywords:** insulin resistance, islet β cell, type 2 diabetes mellitus, gallbladder stone, HbA1c

## Abstract

Background: This study aimed to analyze the correlation between islet β cell function and gallbladder stone (GBS) in newly diagnosed type 2 diabetes mellitus (T2DM) patients. Methods: A total of 438 newly diagnosed T2DM patients in Peking University International Hospital from January 2017 to August 2022 were retrospectively analyzed and divided into a non-GBS group and a GBS group. Results: (1) The homeostasis model assessment of the insulin resistance (HOMA-IR) of the GBS group was higher than that of the non-GBS group (*p* < 0.05), while the homeostasis model assessment of β cell (HOMA-β), disposition index (DI0), and Matsuda index of the GBS group were lower than those of the non-GBS group (all *p* < 0.05). (2) For male patients, HOMA-IR is an independent risk factor for GBS (OR = 2.00, 95% CI:1.03, 3.88, *p* < 0.05), and the Matsuda index value is a protective factor for GBS (OR = 0.76, 95% CI:0.60, 0.96, *p* < 0.05). For female patients, HOMA-IR is an independent risk factor for GBS (OR = 2.80, 95% CI:1.03, 7.58, *p* < 0.05) and the Matsuda index value is a protective factor for GBS (OR = 0.59, 95% CI:0.39, 0.90, *p* < 0.05). (3) For male patients, the area under curve (AUC) for predicting GBS was 0.77 (95% CI 0.67, 0.87), with a specificity of 75.26%, a sensitivity of 80.00%, and an accuracy of 75.64%. For female patients, the AUC for predicting GBS was 0.77 (95% CI 0.63, 0.88), with a specificity of 79.63%, a sensitivity of 71.43%, and an accuracy of 78.69%. Conclusions: Insulin resistance may be an independent risk factor for the incidence of GBS in patients with newly diagnosed T2DM, both male or female, which provides a new clinical basis and research direction for the prevention and treatment of GBS in patients with T2DM. This study has established a predictive model of GBS in T2DM and found it to be accurate, thus representing an effective tool for the early prediction of GBS in patients with T2DM.

## 1. Introduction

Gallbladder stone (GBS) is one of the most common diseases requiring surgery in health care systems. In recent years, with the improvement in people’s living standards and the change in eating habits, the incidence of this disease has increased year by year [1,2]. Results have shown that the prevalence of diabetes mellitus (DM) with GBS is about two times that of normal people, and about 1/3 of GBS patients are found to have DM when biliary tract surgery is performed [3,4]. Elmehdawi et al. [5] found that obese and elderly patients with DM were more likely to suffer from GBS and believed that weight, age, and gender were risk factors for diabetes associated with GBS. Early studies believed that diabetes fat metabolism disorder, insulin resistance, and autonomic neuropathy were high-risk factors for the formation of gallstones associated with diabetes. In recent years, there has been some research progress, such as studies on adiponectin, leptin, metabolic syndrome, cholecystokinin (CCK), and so on. The dyslipidemia in patients with DM is characterized by hypertriglyceridemia, lower levels of high-density lipoprotein cholesterol (HDL-C), and higher levels of low-density lipoprotein cholesterol (LDL-C) [6]. DM is complicated with GBS, and its symptoms are closely related to the decline in HDL and the rise in LDL. In addition, due to insulin resistance or insufficient insulin secretion, DM cannot effectively inhibit fat decomposition, and type 2 diabetes mellitus (T2DM) is often accompanied by obesity, hyperinsulinemia, and dyslipidemia, which ultimately leads to an increase in cholesterol synthesized in the liver, an imbalance in the proportion of cholesterol, bile acid and phospholipid in bile, a saturated cholesterol, and poor water solubility, which increases the risk of forming GBS significantly [7].

The clinical manifestations of GBS are severe pain in the upper abdomen or right upper abdomen and radiation to the right shoulder or back, accompanied by nausea and vomiting. If complicated with acute cholecystitis or pancreatitis, there may even be chills and fever. However, most patients with DM have neuropathy, and their sensitivity to pain is reduced. When they have GBS, they always have no obvious symptoms, and once they get sick, they may suffer from gangrene and perforation of gallbladder, or even severe cholangitis and severe pancreatitis. Therefore, exploring the risk factors of GBS in T2DM patients is of great clinical significance to prevent serious complications of GBS in patients with T2DM.

It has been shown that dyslipidemia and diabetic autonomic neuropathy are the highest risk factors of GBS in T2DM patients. At the same time, some studies have shown that the changes in adiponectin, leptin, and cholecystokinin levels in T2DM patients are also closely related to the incidence of GBS. Recent studies have found that [8] the insulin levels of patients with T2DM and GBS are also significantly higher than in those without GBS. However, due to the limitation of sample size and the fact that the islet β cell secretion levels of newly diagnosed patients will be affected by the duration of DM, previous studies cannot truly reflect the relationship between islet β cell secretion function and GBS in T2DM patients. Therefore, the innovation in our study was to explore the relationship between islet β cell secretion function and GBS in newly diagnosed T2DM patients, in order to minimize the bias caused by medication and the duration of T2DM.

In our study, the relationship between insulin resistance and islet β cell secretion function and the incidence of GBS in patients with newly diagnosed T2DM were analyzed to further clarify the influence of islet β cell function on the incidence of GBS and to provide clinical evidence for the prevention and progression of GBS.

## 2. Materials and Methods

### 2.1. Ethics Statement

This study was approved by the Ethics Committee of Peking University International Hospital, and all methods were performed in accordance with the relevant guidelines and regulations. Due to the retrospective nature of the study, the Ethics Committee of Peking University International Hospital waived the need to obtain informed consent.

### 2.2. Research Subjects

This study retrospectively analyzed 438 newly diagnosed T2DM patients from January 2017 to August 2022 in Peking University International Hospital, including 315 males and 123 females, with an average age of 55.49 ± 13.65 years. All patients met the diagnostic criteria for diabetes issued by the World Health Organization (WHO) in 1999 [9], including: ① typical symptoms of diabetes and random blood glucose ≥ 11.1 mmol/L; ② fasting blood glucose ≥ 7.0 mmol/L; ③ in the oral glucose tolerance test (OGTT), blood glucose ≥ 11.1 mmol/L after taking 75 g glucose for 2 h. If there are no symptoms of diabetes, the above examination should be repeated on another day. If one of the three criteria is then met, the patient is diagnosed as suffering from diabetes and meets the diagnostic criteria for type 2 diabetes according to the clinical classification.

The exclusion criteria of this study were ① patients with type 1 diabetes, gestational diabetes, and other special types of diabetes; ② acute complications of diabetes; ③ patients with severe liver and kidney function diseases; ④ patients who have been diagnosed with GBS in the past; and ⑤ patients with hematological diseases and malignant tumors.

The graphical scheme of the study design is shown below.



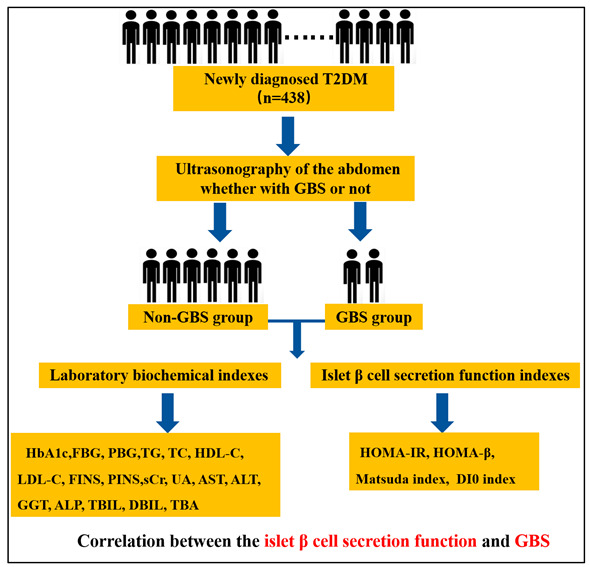



### 2.3. General Conditions and Clinical Data

The data of 438 patients were retrospectively analyzed, and the general conditions of the patients, including sex, age, height, weight, systolic blood pressure (SBP), and diastolic blood pressure (DBP) were recorded. The body mass index (BMI) was calculated and recorded for the patients: BMI (kg/m^2^) = height (kg)/body weight^2^ (m^2^).

### 2.4. Laboratory Biochemical Indexes 

All subjects were patients with T2DM diagnosed in the outpatient department or in the hospital for the first time. All subjects had had an empty stomach for more than 8 h, and the OGTT was performed in the morning of the next day; the fasting biochemical indexes include glycosylated hemoglobin (HbA1c), fasting blood glucose (FBG), triglyceride (TG), total cholesterol (TC), high-density lipoprotein cholesterol (HDL-C), low-density lipoprotein cholesterol (LDL-C), fasting insulin (FINS), serum creatinine (sCr), uric acid (UA), aspartate aminotransferase (AST), alanine aminotransferase (ALT), gamma-glutamyl transpeptidase (GGT), alkaline phosphatase (ALP), total bilirubin (TBIL), direct bilirubin (DBIL), and total bile acid (TBA). A total of 75 g of glucose powder was then dissolved in 300 mL of warm boiled water and this was drunk within 5 min. Starting from the first sip of glucose, postprandial blood glucose (PBG) and postprandial insulin (PINS) were measured after 2 h. Glomerular filtration rates (eGFR) were calculated based on the sCr of the study subjects. All blood tests were performed at the central laboratory of Peking University International Hospital. FBG, UA, sCr, AST, ALT, GGT, ALP, TBIL, DBIL, and TBA were measured using enzyme-based techniques. HbA1c was determined through high-performance liquid chromatography, and the detection instrument model was Japan Dongcao G8 glycosylated hemoglobin analyzer.

### 2.5. Islet β Cell Secretion Function Index Evaluation

According to the measurement results, homeostasis model assessment (HOMA) was used to assess insulin sensitivity and insulin resistance, and the homeostasis model assessment of β cell (HOMA–β) was used to evaluate islet β cell secretion function, using the following calculation formula: HOMA-β = FINS × 20/(FBG − 3.5).

The Matsuda index [10] was used to evaluate insulin sensitivity after glucose load, and the following calculation formula was used: Matsuda index = 10,000/$\sqrt{FINS*FBG*MeanBG*MeanINS}$). 

The homeostasis model assessment of insulin resistance (HOMA-IR) was used to evaluate the degree of insulin resistance, and the calculation formula used was as follows: HOMA-IR = FINS × FBG/22.5. 

The disposition index (DI0) [11] was used to evaluate the basic islet β cell secretion function after adjustment of HOMA-IR, and the calculation formula was as follows: DI0 = HOMA-β/HOMA-IR.

### 2.6. Ultrasonography of the Abdomen

All patients were required to fast for more than 12 h before testing. The following morning, a qualified ultrasound physician used an Echolaser X4 (Esaote SpA, Genoa, Italy) color Doppler ultrasound instrument and performed an abdominal ultrasound on each patient. The status of the gallbladder and bile duct were recorded in detail. 

The patients were divided into two groups according to the presence of GBS: the GBS group (*n* = 39) and the normal group (*n* = 399).

### 2.7. Statistical Methods

SPSS 22.0 software (IBM, Chicago, IL, USA) was used for statistical analysis. The measurement data of the normal distribution is expressed by mean ± standard deviation (x ± s). The data of non-normal distribution is converted to normal distribution via square root (SQRT) and then analyzed, and *t*-test was used to compare the normal distribution data between the two groups. The ratio was used to describe counting data, and this study uses the χ2 test to compare counting data between two groups. This study used an unconditional linear regression model to conduct univariate and multivariate analysis of the risk factors for GBS occurrence, as well as calculated odds ratios (OR) and 95% confidence intervals (CI). This study evaluated the predictive and diagnostic abilities of the models by drawing the receiver operating characteristic curve (ROC). All statistical tests were performed as two-sided tests, and a *p*-value < 0.05 was considered statistically significant.

## 3. Results

### 3.1. Comparison of General Conditions and Biochemical Indexes between the Two Groups

Compared with the normal group, the GBS group was older (*p* < 0.05). The GGT of the GBS group was higher than that of the normal group (*p* < 0.05), while the FINS was higher in the GBS group (*p* < 0.05). The SQRTHOMA-IR of the GBS group was higher than that of the normal group (*p* < 0.05), while the SQRTHOMA-β, SQRTDI0, and SQRTMatsuda of the GBS group were lower than that of the normal group (*p* < 0.05, respectively).There were no significant differences in sex, BMI, SBP, DBP, FBG, PBG, HbA1c, AST, ALT, ALP, TC, TG, LDL-C, HDL-C, TBIL, DBIL, TBA, PINS, eGFR, and UA levels between the two groups (*p* > 0.05, respectively) (Table 1).

### 3.2. Comparison of Islet β Cell Function Index Evaluation between the Two Groups

In male patients, the SQRTHOMA-IR of the GBS group was higher than that of the normal group (*p* < 0.05), while the SQRTHOMA-β, SQRTDI0, and SQRTMatsuda of the GBS group were lower than those of the normal group (*p* < 0.05, respectively).

In female patients, the SQRTHOMA-IR of the GBS group was higher than that of the normal group (*p* < 0.05), while the SQRTHOMA-β, SQRTDI0, and SQRTMatsuda of the GBS group were lower than those of the normal group (*p* < 0.05, respectively) (Figure 1).

### 3.3. Univariate Logistic Regression Analysis of GBS

Taking GBS as the dependent variable, a univariate logistic regression model was established. For male patients, after adjustment for age, BMI, ALT, and eGFR, HOMA-IR is an independent risk factor for GBS (OR = 2.00, 95% CI:1.03, 3.88, *p* < 0.05). The Matsuda index value is a protective factor for GBS (OR = 0.76, 95% CI:0.60, 0.96, *p* < 0.05). For female patients, after adjustment for age, BMI, ALT, and eGFR, HOMA-IR is an independent risk factor for GBS (OR = 2.80, 95% CI:1.03, 7.58, *p* < 0.05). The Matsuda index value is a protective factor for GBS (OR = 0.59, 95% CI:0.39, 0.90, *p* < 0.05) (Figure 2).

### 3.4. Multivariate Predictive Model for Predicting GBS

For male patients, multivariate predictive model for predicting the occurrence of GBS was established with GBS as the dependent variable and sex, age, BMI, HOMA-IR, HOMA-β, DI0, and Matsuda as independent variables. The logistic regression model for predicting the GBS was −3.73510 + 0.06236×age + 0.02119×BMI + 0.37235×HOMA-IR − 0.26797×HOMA-β + 0.33766×DI0-0.32066×Matsuda. The area under curve (AUC) of the model for predicting GBS was 0.77 (95% CI 0.67, 0.87), with the specificity of the model being 75.26%, the sensitivity of the model being 80.00%, and the accuracy of the model being 75.64%.

For female patients, a multivariate predictive model for predicting the occurrence of GBS was established with GBS as the dependent variable and sex, age, BMI, HOMA-IR, HOMA-β, DI0, and Matsuda as independent variables. The logistic regression model for predicting the GBS was 1.62159 + 0.01908×age − 0.15143× HOMA-IR − 0.09614×DI0 − 0.54718× Matsuda. The area under curve (AUC) of the model for predicting GBS was 0.77 (95% CI 0.63, 0.88), with the specificity of the model being 79.63%, the sensitivity of the model being 71.43%, and the accuracy of the model being 78.69% (Figure 3).

## 4. Discussion

GBS is closely related to DM, especially to T2DM. The proportion of patients with T2DM who suffer from GBS is more than 30%; while the incidence of GBS in patients without DM is only 10%, the incidence of GBS with DM is three times that without DM [12,13]. Therefore, the study of the relationship between GBS and T2DM has important clinical significance for the diagnosis and treatment of patients with both GBS and T2DM.

As early as more than 50 years ago, Sampler et al. found in epidemiological studies that the incidence of GBS in Indians with hyperinsulinemia and significant insulin resistance was extremely high. This study suggests that insulin resistance may be related to the incidence of GBS [14]. Scragg et al. also found that the plasma insulin level was closely related to the risk of GBS [15]. In patients with GBS, fasting insulin levels were higher than those in controls, and the insulin level was a risk factor independent of age, BMI, and TG. A study has found that the incidence of GBS is related to metabolic syndrome and the key factor is hyperinsulinemia [16]. Kboi Q et al. found that hyperglycemia, hyperinsulinemia, and hyperlipidemia can reduce the contractility of gallbladder smooth muscle because of various neurotransmitters in the study of mouse gallbladders in vitro [17].

The clinical features of patients with T2DM are mainly hyperinsulinemia and insulin resistance. The mechanism of GBS induced by increasing insulin level is still unclear and may include the following aspects: (1) Insulin may increase the activity of hydroxymethylglutaryl-coenzyme A reductase, which leads to increased cholesterol synthesis and increased hepatic cholesterol secretion [18] to saturate the cholesterol in the bile. And it may promote the formation of GBS. (2) Insulin inhibits the rate-limiting enzyme-7α of bile salt synthesis. The activity of hydroxylase can reduce the bile acid secretion in the bile duct, increase the calcium ion content in the bile, and induce the gallbladder to secrete more mucopolysaccharide substances, which are known as nucleating factors in the bile, breaking the dynamic balance between pro-nucleation and anti-nucleation factors in bile and accelerating the crystallization rate of cholesterol in bile, thereby promoting the formation of GBS [19]. (3) The elevation of insulin can increase the number of LDL receptors, thereby improving the activity of LDL receptors and increasing the amount of LDL-C transferred from the blood circulation to the liver, which may prompt the liver to excrete more cholesterol into the bile and further increase the saturation of cholesterol in the bile [20]. (4) Insulin regulates the contraction of smooth muscle cells by regulating the ion concentration of the intracellular Na^+^ -K^+^ pump, thereby reducing the activity of the gallbladder and causing the formation of GBS. (5) Patients with DM often exhibit abnormal symptoms of autonomic nervous function, which lead to impaired gallbladder contraction function, thickening, and deposition of bile secretion concentration, which is conducive to the formation of GBS [21]. (6) Diabetes causes endocrine disorders, especially the regulation of the lipid metabolism, which is closely related to the incidence of hyperlipidemia and cholesterolemia, especially affecting the secretion of cholesterol, leading to an increase in free cholesterol and fatty acids, thus promoting the formation of stones [22].

Studies have shown that there are many methods currently used clinically to evaluate the islet β cell function and insulin resistance in patients with T2DM, including direct detection of insulin sensitivity and indirect detection of insulin sensitivity. Hyperinsulinemic euglycemic clamp (HEC) technology and the insulin suppression test (IST) were used in the pathogenesis research and new drug evaluation of various metabolic diseases. For population research, it is still appropriate to use indirect detection methods of insulin sensitivity in the fasting state, such as HOMA-IR and HOMA-β. However, for T2DM patients with a long duration and those who have previously used hypoglycemic drugs, such as sulfonylureas and other insulin secretagogue, insulin levels cannot truly reflect the islet β-cell function, and a homeostasis model evaluation may not reflect the degree of insulin resistance in T2DM patients. However, in this study, all the subjects were patients diagnosed with T2DM for the first time, who had not used hypoglycemic therapy before, so that the insulin levels of the subjects, and especially the insulin resistance index calculated by the homeostasis model, can better reflect the islet β-cell secretion function and the degree of insulin resistance.

The results of this study showed that after sex stratification, the HOMA-IR of the GBS group was higher than that of the normal group (*p* < 0.05), while the HOMA-β, DI0, and Matsuda of the GBS group were lower than those of the normal group (*p* < 0.05, respectively). And in univariate regression analysis, for male patients, after adjustment for age, BMI, ALT, and eGFR, HOMA-IR is an independent risk factor for GBS (OR = 2.00, 95% CI:1.03, 3.88, *p* < 0.05). The Matsuda index value is a protective factor for GBS (OR = 0.76, 95% CI:0.60, 0.96, *p* < 0.05). For female patients, after adjustment for age, BMI, ALT, and eGFR, HOMA-IR is an independent risk factor for GBS (OR = 2.80, 95% CI:1.03, 7.58, *p* < 0.05). The Matsuda index value is a protective factor for GBS (OR = 0.59, 95% CI:0.39, 0.90, *p* < 0.05). This result is consistent with previous research results [23,24]. In a study that included 4125 Korean postmenopausal women [24], the results showed that in a multiple logistic regression analysis, LDL-C was an independent factor of GBS in premenopausal women and HOMA-IR was significantly associated with the occurrence of the GBS. However, previous studies only analyzed the correlation between insulin levels and GBS, while we further adopted more accurate indicators that reflect islet β cell secretion levels and insulin resistance indexes, which can better verify that the severity of insulin resistance in T2DM patients is an independent risk factor for the incidence of GBS, and that the degree of islet β cell secretion is closely related to the incidence of GBS. 

In recent studies, there are also many reports about the related risk factors for GBS in T2DM patients [25]. Physiological factors, especially sex and age, have a great influence on the incidence of GBS in patients with T2DM. With the increase in age, the function of various organs in the human body exhibits an aging failure trend, which directly leads to the decrease in cholesterol transformation into bile acid. The decrease in cholesterol decomposition increases the concentration of cholesterol in the total cholesterol pool and thus increases the cholesterol excretion in the biliary tract. The results of this study also confirm this point, as the average age in the GBS group was higher than that in the normal group (*p* < 0.05). 

Estrogen is the main factor responsible for the different incidence rates of GBS between men and women. Studies have shown that the level of estrogen in women can affect the secretion of bile acids. The higher the estrogen level, the lower the amount of bile acids secretion, which results in an increase in the incidence of GBS. However, the results of this study have shown that compared with the normal group, the proportion of female patients in the GBS group was a little higher, without a significant difference. This difference between studies may be the result of the limited sample size. Also, the effect of sex difference on GBS is mainly due to the difference in estrogen levels. The women included in a previous study [26] were young, so there was a significant correlation between being female and GBS. However, the average age of the women in our study was more than 50 years old, most of them were postmenopausal women, and the estrogen levels in postmenopausal women are significantly lower than in younger women. Therefore, our results could not reflect the effect of estrogen levels on GBS in patients with T2DM. 

Our results showed that the levels of GGT in the GBS group were slightly higher than those in the non-GBS group (*p* < 0.05), but there were no significant differences in the levels of TBIL, DBIL, ALP, and TBA between the two groups (all *p* > 0.05). In the further univariate logistic regression analysis, none of the above indexes reflecting liver and gallbladder functions were risk factors for GBS in T2DM patients (*p* > 0.05), which may be because these indexes only reflect the physiological functions of the liver and gallbladder and do not directly affect cholesterol excretion, so there is no apparent correlation between them and the incidence of GBS.

In this study, a multivariate logistic regression model was established for predicting the incidence of GBS. For male patients, the multivariate predictive model was established with GBS as the dependent variable and sex, age, BMI, SQRTHOMA-IR, SQRTHOMA-β, SQRTDI0, and SQRTMatsuda as independent variables. The equation of the logistic regression model was logit (GBS) = −3.73510 + 0.06236*age + 0.02119*BMI + 0.37235*HOMA-IR − 0.26797*HOMA-β + 0.33766*DI0-0.32066*Matsuda. And the AUC of the model for male patients is 0.77 (95% CI 0.67, 0.87), with a specificity of 75.26%, a sensitivity of 80.00%, and an accuracy of 75.64%. For female patients, the multivariate predictive model was established with GBS as the dependent variable and sex, age, BMI, SQRTHOMA-IR, SQRTHOMA-β, SQRTDI0, and SQRTMatsuda as independent variables. The logistic regression equation of the model was logit (GBS) = 1.62159 + 0.01908*age − 0.15143* HOMA-IR − 0.09614*DI0 − 0.54718* Matsuda. The AUC of the model for female patients is 0.77 (95% CI 0.63, 0.88), with a specificity of 79.63%, a sensitivity of 71.43%, and an accuracy of 78.69%.

This study not only identified the risk factors for GBS in T2DM patients, but also further established a predictive model for GBS in T2DM patients, which has a certain predictive value. This model has not been investigated in previous studies. Through the establishment of this model, we can evaluate the risk of GBS occurrence in newly diagnosed T2DM patients. For T2DM patients with a higher risk, further diagnosis and evaluation of the necessity of surgery for GBS should be actively carried out to prevent the deterioration of GBS.

There are still some limitations to this study. First, one of the limitations of our research is that it is a retrospective study, and we had limited control over the quality of the data used for analysis. Secondly, some biochemical indexes were tested only once, which may lead to variation. In the future, studies with larger samples need to be carried out. In addition, this study did not reveal the pathophysiological mechanism of the close relationship between islet β cell secretion and GBS. In the future, more animal studies are needed to better understand the pathophysiological basis for the close relationship between islet β cell secretion and GBS. Finally, the sample size of this study was small, especially the sample size of the GBS group, which was only 39 cases. In order to avoid the influence of hypoglycemic treatment and the diabetes duration on insulin function and insulin resistance as much as possible, the subjects of this study were newly diagnosed T2DM patients. Although the sample size was limited, the sample size ratio of the non-GBS group and GBS group was 10:1, which was in line with statistical requirements. At the time of writing, there were not many studies on the correlation between insulin function, insulin resistance indexes, and GBS in newly diagnosed T2DM patients. At that point, our study had the largest sample size, but this was only a preliminary exploration. In future research, we will further expand the sample size to better reveal the correlation between pancreatic function, insulin resistance indicators, and GBS.

## 5. Conclusions

Insulin resistance may be an independent risk factor for the incidence of GBS in patients with newly diagnosed T2DM, both male and female, which provides a new clinical basis and research direction for the prevention and treatment of GBS in patients with T2DM. This study has established a predictive model of GBS in T2DM and found it to be accurate, thus representing an effective tool for the early prediction of GBS in patients with T2DM.

## Figures and Tables

**Figure 1 biomedicines-11-02840-f001:**
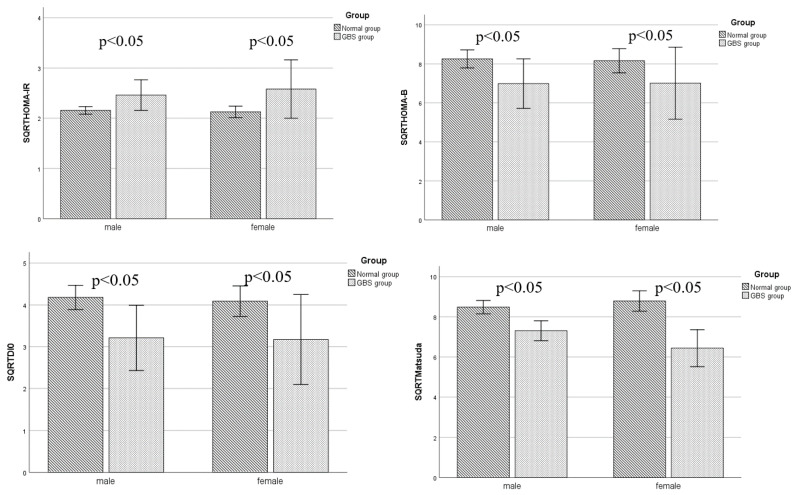
Comparison of islet β cell function index evaluation between the two groups. In both male and female patients, the SQRTHOMA-IR of the GBS group was higher than that of the normal group (all *p* < 0.05), while the SQRTHOMA-β, SQRTDI0, and SQRTMatsuda of the GBS group were lower than those of the normal group (all *p* < 0.05).

**Figure 2 biomedicines-11-02840-f002:**
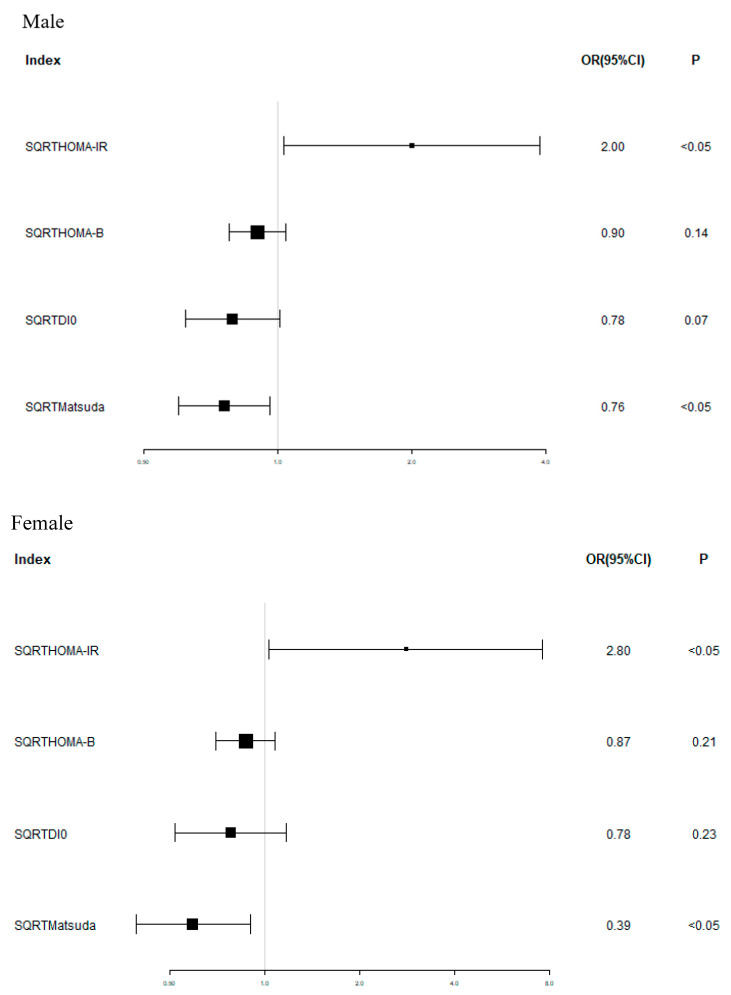
Univariate logistic regression analysis of GBS. In both male and female patients, HOMA-IR is an independent risk factor for GBS (male: OR = 2.00, 95% CI:1.03, 3.88, *p* < 0.05; female: OR = 2.80, 95% CI:1.03, 7.58, *p* < 0.05), and Matsuda index value is a protective factor for GBS (male: OR = 0.76, 95% CI:0.60, 0.96, *p* < 0.05; female: OR = 0.59, 95% CI:0.39, 0.90, *p* < 0.05).

**Figure 3 biomedicines-11-02840-f003:**
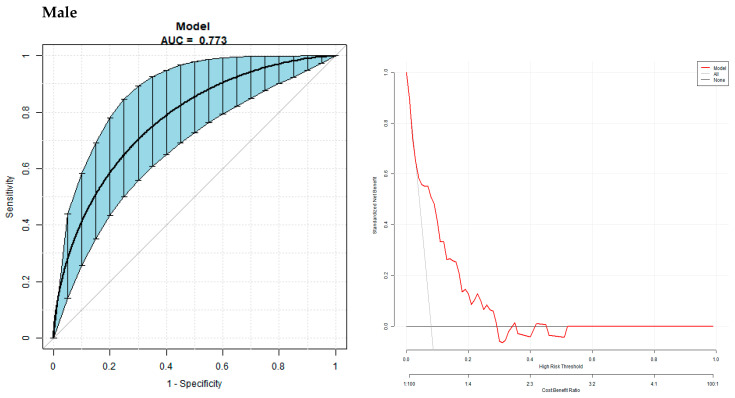
Multivariate predictive model for the risk of GBS. The AUC of the model for predicting the occurrence of GBS in male patients was 0.77 (95% CI 0.67, 0.87), with a specificity of 75.26%, a sensitivity of 80.00%, and an accuracy of 75.64%. The AUC of the model for predicting the occurrence of GBS in female patients was 0.77 (95% CI 0.63, 0.88), with a specificity of 79.63%, a sensitivity of 71.43%, and an accuracy of 78.69%.

**Table 1 biomedicines-11-02840-t001:** Comparison of general conditions and biochemical indexes between the two groups.

Index	Non-GBS Group	GBS Group	t(χ2)	*p*
	(*n* = 399)	(*n* = 39)		
Age (years)	47.66 ± 14.83	57.13 ± 15.60	−3.83	<0.05
Sex (female%)	109 (27.32%)	14 (35.90%)	1.29	0.26
BMI (kg/m^2^)	26.45 ± 4.37	27.29 ± 4.70	−1.12	0.26
SBP (mmHg)	131.28 ± 17.51	133.90 ± 14.10	−0.91	0.37
DBP (mmHg)	81.05 ± 11.87	80.23 ± 10.38	0.42	0.68
AST (U/L)	26.73 ± 9.66	23.72 ± 17.48	1.94	0.08
ALT (U/L)	27.50 ± 10.04	28.96 ± 16.84	−0.99	0.45
GGT (U/L)	32.06 ± 21.48	39.35 ± 32.17	−2.15	<0.05
ALP (U/L)	55.58 ± 14.96	50.85 ± 13.96	1.46	0.41
TBIL (umol/L)	19.35 ± 7.09	23.03 ± 6.54	−1.28	0.65
DBIL (umol/L)	4.39 ± 0.43	4.40 ± 0.54	−0.43	0.85
TBA (umol/L)	4.86 ± 1.09	4.07 ± 1.15	0.78	0.34
TC (mmol/L)	4.60 ± 1.29	4.53 ± 1.57	0.29	0.77
TG (mmol/L)	2.32 ± 1.71	2.15 ± 2.71	1.48	0.14
LDL-C (mmol/L)	2.83 ± 0.92	2.71 ± 1.35	0.69	0.49
HDL-C (mmol/L)	0.94 ± 0.22	0.93 ± 0.22	0.15	0.88
UA (umol/L)	335.90 ± 89.21	324.76 ± 80.23	0.96	0.65
eGFR	103.27 ± 18.69	108.54 ± 18.87	−1.88	0.07
HbA1c (%)	9.47 ± 2.43	10.01 ± 2.32	−1.06	0.29
FBG (mmol/L)	9.26 ± 3.80	9.79 ± 3.83	−0.78	0.44
PBG (mmol/L)	12.23 ± 5.59	11.98 ± 4.18	0.26	0.79
FINS (uU/mL)	14.65 ± 16.06	19.75 ± 7.27	−2.02	<0.05
PINS (uU/mL)	42.76 ± 40.79	46.02 ± 63.28	−1.20	0.24
SQRTHOMA-IR	2.15 ± 0.64	2.50 ± 0.83	−3.23	<0.05
SQRTHOMA-β	8.23 ± 3.81	7.00 ± 3.08	2.33	<0.05
SQRTDI0	4.15 ± 2.36	3.20 ± 1.86	2.45	<0.05
SQRTMatsuda	8.57 ± 2.83	7.00 ± 1.40	3.42	<0.05

Note: BMI is the abbreviation for body mass index, SBP is the abbreviation for systolic blood pressure, DBP is the abbreviation for diastolic blood pressure, FBG is the abbreviation for fasting blood glucose, PBG is the abbreviation for postprandial blood glucose, FINS is the abbreviation for fasting insulin, PINS is the abbreviation for postprandial insulin, HbA1c is the abbreviation for glycosylated hemoglobin, eGFR is the abbreviation for glomerular filtration rates, UA is the abbreviation for uric acid, UACR is the abbreviation for urinary albuminuria creatinine ratio, TC is the abbreviation for total cholesterol, TG is the abbreviation for triglycerides, LDL-C is the abbreviation for low-density lipoprotein cholesterol, HDL-C is the abbreviation for high-density lipoprotein cholesterol, AST is the abbreviation for aspartate aminotransferase, ALT is the abbreviation for alanine aminotransferase, GGT is the abbreviation for gamma-glutamyl transpeptidase, ALP is the abbreviation for alkaline phosphatase, TBIL is the abbreviation for total bilirubin, DBIL is the abbreviation for direct bilirubin, TBA is the abbreviation for total bile acid, SQRTHOMA-IR is the abbreviation for square root of homeostasis model assessment insulin resistance, SQRTHOMA-β is the abbreviation for square root of homeostasis model assessment β, SQRTDI0 is the abbreviation for square root of disposition index, SQRTMatsuda is the abbreviation for square root of Matsuda.

## Data Availability

The datasets generated and analyzed during the current study are not publicly available due to limitations of ethical approval involving patient data and anonymity but are available from the corresponding author on reasonable request.

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
