# Peer review of "The Correlation between Islet β Cell Secretion Function and Gallbladder Stone Disease: A Retrospective Study Based on Chinese Patients with Newly Diagnosed Type 2 Diabetes Mellitus"

_biomedicines, 2023, doi:10.3390/biomedicines11102840_

Round 1

Reviewer 1 Report

The manuscript was prepared very well. The introduction section justifies the purpose of the study. I congratulate the authors for the preparation of the manuscript

I would like to congratulate the authors for the structure of the manuscript and all the research carried out. It is highly publishable. However, there are some concerns, in part important, so the review articles need revision, see below.

Introduction

-        Why is this study considered relevant?

-        Why is this study necessary?

Methods and Results

-        It is one of the strong parts of the manuscript, these excellently described

Discussion

·       Include a section on strengths / limitations.

·       What mechanisms of action support these findings?

·       What does this article contribute to, the authors should make their own assessment and include their own discussion of the results shown in the manuscript?

Conclusion

In the Conclusion section, state the most important outcome of your work. Do not simply summarize the points already made in the body — instead, interpret your findings at a higher level of abstraction. Show whether, or to what extent, you have succeeded in addressing the need stated in the Introduction (or objectives).

You should check typo errors and some expressions

Author Response

Replies to Reviewer

Comments

  1. Introduction
  • Why is this study considered relevant?
  • Why is this study necessary?

Response: Thank you for your insightful question. Because there are many common risk factors in diabetes and GBS, such as obesity, age, dyslipidemia and insulin resistance, which suggests that there may be some internal correlation between the two diseases. In addition, many patients with diabetes have neuropathy, and their sensitivity to pain is reduced. When they have GBS, they often have no typical symptoms, which may cause severe anxiety and severe pancreatitis. Therefore, we hope to explore the risk factors of GBS in diabetes patients, so that they can be prevented earlier. We have revised the introduction and explained the study necessary in the red in revised manuscript.

  1. Discussion
  • Include a section on strengths / limitations.
  • What mechanisms of action support these findings?
  • What does this article contribute to, the authors should make their own assessment and include their own discussion of the results shown in the manuscript?

Response: Thank you for your insightful question. We have revised the discussion part and added the limitations and strengths, the mechanisms of action and the article contribution in red in revised manuscript.

  1. Conclusion

In the Conclusion section, state the most important outcome of your work. Do not simply summarize the points already made in the body — instead, interpret your findings at a higher level of abstraction. Show whether, or to what extent, you have succeeded in addressing the need stated in the Introduction (or objectives).

Response: Thank you for your insightful suggestion. We have revised the conclusion part in red in revised manuscript.

Reviewer 2 Report

Current study investigated the correlation between islet β cell function and gallbladder stone (GBS) in newly diagnosed type 2 diabetes mellitus (T2DM) patients. I like to give the following comments.

1.      Abstract seems unclear and it needs to rephrase in addition.

2.      Gallbladder stone (GBS) associated with metabolic disorders must introduce in detail. Additionally, error in spelling of GBS on line 28 was observed.

3.      GBS as highest risk factor in T2DM patients needs reference(s) to support.

4.      Rationale for targeting the incidence of GBS in patients with newly diagnosed T2DM was unknown. Please introduce it in clear.

5.      Ethical data for this analysis in hospital such as IRB was not indicated.

6.      Kits used for assay of blood parameters were not mentioned. Why?

7.      Matsuda index or Disposition index (DI0) needs the reference(s) to support.

8.      The difference may be due to the low sample size (n=39) in GBS group. How to rule out this possibility?

9.      Biomarker(s) of GBS seems ignored in current study. Why?

10.  The newly diagnosed T2DM patients were not conducted in conclusion. Why?

Author Response

Replies to Reviewer

Comments

  1. Abstract seems unclear and it needs to rephrase in addition.

Response: Thank you for your insightful question. We have revised the abstract in the red in revised manuscript.

  1. Gallbladder stone (GBS) associated with metabolic disorders must introduce in detail. Additionally, error in spelling of GBS on line 28 was observed.

Response: Thank you for your insightful suggestion. We have revised the introduction and added the GBS and metabolic disorders in the introduction in the red in revised manuscript.

  1. GBS as highest risk factor in T2DM patients needs reference(s) to support.

Response: Thank you for your insightful suggestion. We have added the references in the introduction part in revised manuscript.

  1. Rationale for targeting the incidence of GBS in patients with newly diagnosed T2DM was unknown. Please introduce it in clear.

Response: Thank you for your insightful suggestion. Due to the limitation of sample size and the fact that the islet β cell secretion levels of newly diagnosed patients will be affected by the duration of DM, the previous studies cannot truly reflect the relationship between islet β cell secretion function and GBS in T2DM patients. Therefore, it is necessary to explore the relationship between islet β cell secretion function and GBS in newly diagnosed T2DM patients, so as to minimize the bias caused by medication and the duration of T2DM.We have added the necessity in the introduction part in revised manuscript.

  1. Ethical data for this analysis in hospital such as IRB was not indicated.

Response: Thank you for your insightful suggestion. We have added the IRB in method part in red in the revised manuscript.

  1. Kits used for assay of blood parameters were not mentioned. Why?

Response: Thank you for your insightful suggestion. We have added the information in the method part in red in the revised manuscript.

  1. Matsuda index or Disposition index (DI0) needs the reference(s) to support.

Response: Thank you for your insightful suggestion. We have added the references in the revised manuscript.

  1. The difference may be due to the low sample size (n=39) in GBS group. How to rule out this possibility?

Response: Thank you for your insightful suggestion. The sample size of this study was small, especially the sample size of GBS group was only 39 cases. In order to avoid the influence of hypoglycemic treatment and the diabetes duration on insulin function and insulin resistance as far as possible, the subjects of this study were newly diagnosed T2DM patients. Although the sample size of patients was limited, but the sample size ratio of non-GBS group and GBS group was 10:1, which was in line with statistical requirements. There were not many studies on the correlation between insulin function, insulin resistance indexes, and GBS in newly diagnosed T2DM patients. Our study currently has the largest sample size, but this was only a preliminary exploration. In future research, we will further expand the sample size to better reveal the correlation between pancreatic function, insulin resistance indicators and GBS. We have added the limitations in red in the revised manuscript.

  1. Biomarker(s) of GBS seems ignored in current study. Why?

Response: Thank you for your insightful suggestion. This study has conducted analysis of biochemical markers related to GBS, including ALT, AST, GGT, TBIL, DBIL, and TBA. It can be seen in Table 1, but the results only showed an increase in GGT in the GBS group, and no differences were observed in other indicators. In the discussion part, we have also compared and analyzed previous studies, and this part has been highlighted in red in the revised manuscript.

  1. The newly diagnosed T2DM patients were not conducted in conclusion. Why?

Response: Thank you for your insightful suggestion. In order to avoid the influence of hypoglycemic treatment and the diabetes duration on insulin function and insulin resistance as far as possible, the subjects of this study were newly diagnosed T2DM patients. And we have revised the conclusion in red in the manuscript.

Reviewer 3 Report

The Abstract should be better summarized.

Lines 34-51 should be implemented.

The novelty character of paper should be better marked.

A graphical scheme of study design should be inserted.

Introductory lines should be inserted in section Results to better introduce different type of results.

Results in Figures 1 and 2 should be better described.

Results on multivariate predictive model for the risk of GBS should be better described.

Conclusion should be implemented with limits, advantages and future directions.

Author Response

Replies to Reviewer

Comments

  1. The Abstract should be better summarized.

Response: Thank you for your insightful question. We have revised the abstract in the red in revised manuscript.

  1. Lines 34-51 should be implemented.

Response: Thank you for your insightful question. We have revised the introduction in the red in revised manuscript.

  1. The novelty character of paper should be better marked.

Response: Thank you for your insightful suggestion. The novelty of this study is the research subjects are all newly diagnosed T2DM patients. We have added the novelty character in red in the revised manuscript.

  1. A graphical scheme of study design should be inserted.

Response: Thank you for your insightful suggestion. We have added the graphical scheme of study design in the Method in the revised manuscript.

  1. Introductory lines should be inserted in section Results to better introduce different type of results.

Response: Thank you for your insightful suggestion. We have added the introductory lines in Results in the revised manuscript.

  1. Results in Figures 1 and 2 should be better described.

Response: Thank you for your insightful suggestion. We have revised the description of Figure 1 and Figure 2 in red in the revised manuscript.

  1. Results on multivariate predictive model for the risk of GBS should be better described.

Response: Thank you for your insightful suggestion. We have revised the description of Results on multivariate predictive model in red in the revised manuscript.

  1. Conclusion should be implemented with limits, advantages and future directions.

Response: Thank you for your insightful suggestion. We have added the limits, advantages and future directions in the Discussion in red in the revised manuscript.
